# RECEDING NEURON IMPORTANCES FOR STRUCTURED PRUNING

## ABSTRACT

Structured pruning efficiently compresses networks by identifying and removing unimportant neurons. While this can be elegantly achieved by applying sparsity-inducing regularisation on BatchNorm parameters, an L1 penalty would shrink all scaling factors rather than just those of superfluous neurons. To tackle this issue, we introduce a simple BatchNorm variation with bounded scaling parameters, based on which we design a novel regularisation term that suppresses only neurons with low importance. Under our method, the weights of unnecessary neurons effectively recede, producing a polarised bimodal distribution of importances. We show that neural networks trained this way can be pruned to a larger extent and with less deterioration. We one-shot prune VGG and ResNet architectures at different ratios on CIFAR and ImageNet datasets. In the case of VGG-style networks, our method significantly outperforms existing approaches particularly under severe pruning. Source code is available at: `https://anonymous.4open.science/r/receding-neuron-importances-40C9`.

## 1 INTRODUCTION

Modern deep neural network architectures (Simonyan & Zisserman, 2014; He et al., 2016) achieve state-of-the-art performance but require significant computational resources which makes their deployment onto edge devices difficult. Even though it has been shown that it is possible to train less over-parametrised models from scratch and obtain a similar performance (Frankle & Carbin, 2018), it remains a non-trivial task to actually find such a winning subnetwork.

In this work, we focus on structured one-shot pruning as a means to network compression which is typically composed of three stages – i) training a large model to convergence, ii) removing parameters with low importance, and iii) fine-tuning the remaining network. Unstructured pruning, which works on a weight level (Yang et al., 2019; Frankle & Carbin, 2018), can remove a much higher number of parameters but produces sparse weight matrices which cannot be efficiently utilised without specialised hardware (Han et al., 2016). In contrast, by removing entire neurons structured pruning finds efficient structures akin to an implicit architecture search (Liu et al., 2018).

Structured pruning methods attribute an importance score to each neuron, which enables their ranking and ultimately the decision of which to dispose (Li et al., 2016; Molchanov et al., 2016a). To this end, BatchNorm layers (Ioffe & Szegedy, 2015) become very appealing as they explicitly learn parameters which uniformly scale the outputs of each neuron. This scaling parameter can be used as a proxy for the importance the model attributes to a neuron, as a value of zero would effectively suppress an output. Furthermore, one can regularise these layers to obtain neuron level sparsity whilst maintaining classification performance (Liu et al., 2017; Zhuang et al., 2020). Such methods typically define neuron importance as the absolute value of its scaling parameter, an approach which limits the design of regularisers. Because the measure is only half-bounded one cannot easily define levels of importance without looking at the overall distribution - making it difficult to target specific neurons. An example is the L1 regulariser (Liu et al., 2017) which shrinks all parameters with a constant gradient, even ones with high importance. Ideally, one would design a regulariser which creates sparsity by only shrinking unimportant neurons, leaving the others untouched.

In this work, we create such a regulariser and show it outperforms existing approaches at a rate that increases with the amount of neurons pruned. Our contributions are two-fold: we first introduce a simple variation of BatchNorm, which linearly transforms channels using bounded scalers. This

layer maintains the same performance as the original, while offering a bounded importance score for neurons. Building on this measure we then define a novel regularisation, focused on shrinking only neurons with lesser weight, by having its gradient decay exponentially for higher importances. Our method significantly outperforms related approaches for VGG models, and we show that severe degradation can be attributed to over-pruning early layers of the network.

## 2  RELATED WORK

Neural network compression through pruning is most commonly divided into structured and unstructured approaches. Unstructured pruning has gained a lot of attention in recent years (Frankle & Carbin, 2018) as it challenges conventional wisdom over the role of over-parametrisation and weight initialisation in the optimisation of deep neural networks (Frankle et al., 2020). While these methods can achieve superior theoretical compression rates (Renda et al., 2020), their use remains impractical without specialised hardware that can take advantage of sparsity (Han et al., 2016). Structured pruning on the other hand removes entire neurons from an architecture thus achieving real memory and computational efficiencies (Liu et al., 2018).

At the heart of structured pruning lies the task of identifying unimportant neurons to remove from a network. Quantifying importance can be based on numerous criteria including filter norms (Li et al., 2016; He et al., 2018), reconstruction errors (He et al., 2017; Luo et al., 2017; Molchanov et al., 2016b; Yu et al., 2018), redundancy (He et al., 2019; Suau et al., 2020; Wang et al., 2018) and BatchNorm parameters (Liu et al., 2017; Zhuang et al., 2020). Our work belongs in the latter category as we focus on deriving an importance score solely based on channel scaling parameters.

In addition to defining importance measures, one can add regularisation during training to nudge networks into utilising their capacity more sparingly. Most relevant to our work are methods which apply sparsity regularisations on BatchNorm parameters (Liu et al., 2017; Zhuang et al., 2020). The most popular approach is Network Slimming (Liu et al., 2017), which constrains the BatchNorm scaling parameters using the L1 penalty. A drawback of this method is that it shrinks all parameters with an equal gradient irrespective of their importance. This issue is also addressed by Zhuang et al. (2020) who propose a regulariser that explicitly maximises the polarisation of the BatchNorm scaler distribution. While the method effectively increases the margin between important and unimportant neurons, it does so by both shrinking and expanding weights. Another method designed for non-linear shrinking is Yang et al. (2019), who propose the ratio of the L1 and L2 norms as a sparsity regulariser. Even though this method is not based on BatchNorm, but is targeted at filter weights, we include it in our comparison as it has a similar motivation to our work.

In terms of pruning setup the most popular method is one-shot pruning (Li et al., 2016; Zhuang et al., 2020; Yang et al., 2019) where all desired neurons are removed at once and the remaining network is fine-tuned. Iterative approaches (Han et al., 2015) periodically prune and fine-tune until a target ratio is met. Recent works aim to eliminate the need of fine-tuning (Chen et al., 2021) altogether or prune models after intialisation in a data-free manner (Lee et al., 2018; Wang et al., 2020).

## 3  SIGMOID BATCHNORM

We introduce a variation of BatchNorm, which uses a single learnable parameter per channel and offers a bounded importance score for filters. In its original formulation, BatchNorm (Ioffe & Szegedy, 2015) first normalises each input channel $x$ using batch statistics mean $\mu_B$ and standard deviation $\sigma_B$, then applies an affine transformation using learnable parameters $\gamma$ and $\beta$.

$$BN(x, \gamma, \beta) = \gamma \frac{x - \mu_B}{\sqrt{\sigma_B^2 + \epsilon}} + \beta \tag{1}$$

While BatchNorm has become ubiquitous in Deep Learning, the reasons behind its effectiveness are not fully understood (Santurkar et al., 2018). The proliferation of BatchNorm variations (Ba et al., 2016; Wu & He, 2018; Ulyanov et al., 2016) suggests its benefits arise from normalising activations rather than the affine transformation following it. We empirically show that the transformation can be replaced to be linear without loss of performance.

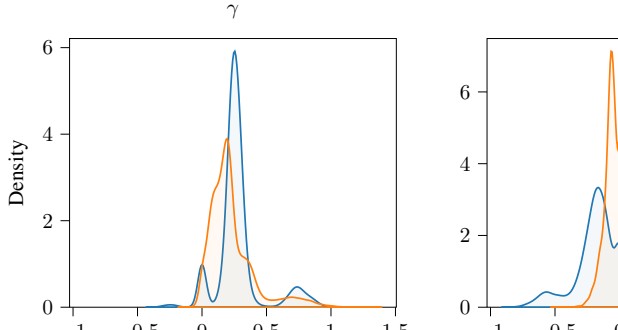

Figure 1: Distribution of parameters across all BatchNorm layers of pre-trained models available in PyTorch (Paszke et al., 2017). The scale $\gamma$ stays in the interval $[0, 1]$, the offset $\beta$ remains close to 0.

We look at the empirical distributions of BatchNorm parameters to understand the effects of the affine transformation. While the scale $\gamma$ and offset $\beta$ are unbounded, in practice, these will be suppressed due to weight decay. We can observe this in pre-trained models: Figure 1 shows the parameter distributions for all BatchNorm layers of a VGG-19 (Simonyan & Zisserman, 2014) and ResNet-50 (He et al., 2016) model. The BatchNorm parameters in both networks are well aligned as almost all $\gamma \in [0, 1]$, while a large proportion of $\beta$ are concentrated around 0.

Because BatchNorm operates on a channel level, it is of particular interest to structured pruning, as it offers a natural place to look for measures that quantify filter importance. In line with our observation about the distribution of $\beta$, previous approaches (Liu et al., 2017; Zhuang et al., 2020) define filter importance as the magnitude of $\gamma$ and ignore the offset. While this has proven to work in practice, one could argue it would be desirable to construct an importance score that incorporates all the information available. Additionally, the utility of such a measure could be greatly improved if it would be bounded - offering a better understanding by knowing the minimum and maximum importance a filter could have.

Given the aforementioned considerations, we propose a simple alteration to BatchNorm, termed $\sigma BN$, which keeps the normalisation scheme, but changes the channel transformation to use a bounded scaling parameter. We remove the offset $\beta$ and bound the scale parameter by applying the sigmoid function to $\gamma$ before multiplication:

$$\sigma BN(x, \gamma) = \sigma(\gamma) \frac{x - \mu_B}{\sqrt{\sigma_B^2 + \epsilon}}, \qquad \text{where} \quad \sigma(\gamma) = \frac{1}{1 + e^{-\gamma}} \tag{2}$$

In Table 1 we see the performance of vanilla BatchNorm (BN) compared with our variation $\sigma$BN. Our variation performs comparable for VGG-16 and ResNet-56 on both CIFAR datasets. For completeness we also add the performance of BN and $\sigma$BN with and without the bias term $\beta$. We can observe that the presence or absence of $\beta$ does not have a profound impact on accuracy. With $\sigma$BN, we now have a normalisation layer that uses a single, bounded parameter to quantify the importance of a filter. We will use this property to decide which filters to suppress and ultimately prune.

| CIFAR10 | BN-$\beta$ | BN | $\sigma$BN | $\sigma$BN+$\beta$ |
|---|---|---|---|---|
| VGG-16 | 93.58 | 93.57 | 93.48 | 93.35 |
| ResNet-56 | 92.87 | 93.34 | 93.11 | 93.33 |

| CIFAR100 | BN-$\beta$ | BN | $\sigma$BN | $\sigma$BN+$\beta$ |
|---|---|---|---|---|
| VGG-16 | 72.81 | 72.99 | 72.75 | 73.12 |
| ResNet-56 | 70.70 | 70.94 | 71.27 | 71.12 |

Table 1: No significant difference in classification accuracy can be seen between any BatchNorm variation, including toggles of the bias term for vanilla BN and $\sigma$BN. Results are averaged over three runs and have standard deviation of $\approx$0.2. Normalisation layers are trained without regularisation.

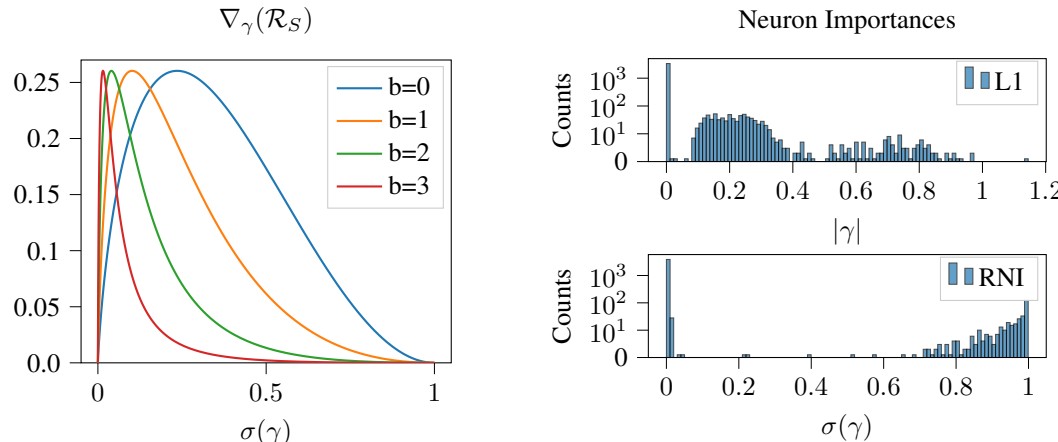

Figure 2: Left: The gradient of the RNI loss with respect to $\gamma$ for different hyper-parameter choices. $b$ shifts the range of neuron importances which the regulariser will affect. Right: Histograms of resulting neuron importances after using L1 or RNI regularisation.

## 4 RECEDING NEURON IMPORTANCE

Structured pruning aims to compress a network by identifying and removing unimportant neurons. We define $\theta$ to be the parameters of a neural network and refer to the subset associated with Batch-Norm as $\theta^{BN}$. Let $I(\theta_i)$ be a measure of importance for neuron $i$, which can be computed from any model parameters associated with that neuron, such as its filters in convolutional layers (Li et al., 2016; Yang et al., 2019). In BatchNorm based pruning, $I$ is derived only from $\theta^{BN}$, and is commonly defined as $I(\theta_i^{BN}) = |\gamma_i|$ (Liu et al., 2017; Zhuang et al., 2020). In this paper we use the measure induced by our $\sigma BN$ layer such that $I(\theta_i^{BN}) = \sigma(\gamma_i)$. As in previous work, we want to solve the following risk minimisation problem:

$$\min_{\theta} \frac{1}{N}\mathcal{L}(\theta, X) + \lambda\mathcal{R}(\theta) + \lambda_S\mathcal{R}_S(\theta^{BN}) \tag{3}$$

where $X = \{x_i, y_i\}_{i=1}^N$ is a labelled dataset with $N$ training samples, and $\lambda, \lambda_S$ are scalar weightings for the regularisation terms $R$ and $R_S$. $\mathcal{R}(\theta)$ is a regularisation against over-fitting, such as weight decay, and is usually applied over all network parameters, including normalisation layers. $R_S$ is targeting only $\theta^{BN}$ such that $I$ exhibits the following property during pruning:

$$\mathcal{L}(\theta\backslash\{\theta_i\}, X) > \mathcal{L}(\theta\backslash\{\theta_j\}, X), \qquad \text{if} \quad I(\theta_i) < I(\theta_j) \tag{4}$$

Such a property creates an ordering which ensures that removing a neuron $\theta_i$ with lower importance will have less impact on model performance, than removing a more important $\theta_j$. We will assess the quality of such an ordering by looking at the model performance after pruning and fine-tuning. However, an ordering is only partially evaluated when removing a single subset of neurons - since the order of the remaining neurons is not taken into account. Therefore, for a comprehensive estimate of the global ordering quality, it is necessary to prune a model at multiple thresholds.

Generally we want $R_S$ to induce sparsity over $\theta^{BN}$ such that outputs of unimportant neurons are entirely suppressed during training. Pruning such a network would result in less damage and consequently a quicker recovery during fine-tuning. The typical choice of $\mathcal{R}_s$ is the L1 norm, which shrinks all scaling parameters with a constant gradient. Such a regularisation strategy simultaneously subdues both important and unimportant neurons with equal weighting. From an optimisation perspective, L1 will create a bias such that at convergence, any equilibrium would require the classification loss to negate the constant gradient of the regulariser. Such a bias might lead to sub-optimal solutions, and can be mitigated by having the regularisation reduce its strength once a suitable, sparse distribution of importances has been found. We show that a better regularisation term can be

designed by focusing on suppressing unimportant neurons, whilst leaving important ones untouched. Without an idea of bounds for $I(\theta^{BN})$ it is difficult to decide on a threshold between important and unimportant neurons. Using the importance measure from our $\sigma BN$ layers we propose a regularisation loss whose strength decays exponentially as $I(\theta^{BN})$ reaches its maximum. We introduce the Receding Neuron Importance regularisation (RNI) on batch normalisation parameters $\gamma$:

$$\mathcal{R}_s(\gamma, b) = \sigma(\gamma + b) \cdot (1 - \log(\sigma(\gamma + b))), \qquad \text{with} \quad \nabla_{\sigma(\gamma+b)}(\mathcal{R}_s) = -\log(\sigma(\gamma + b))$$
$$\text{and} \quad \nabla_\gamma(\mathcal{R}_s) = -\log(\sigma(\gamma + b))\sigma(\gamma + b)(1 - \sigma(\gamma + b)) \tag{5}$$

The hyper-parameter $b$ controls the range defining which neuron importances to target. Its effects can be visualised in Figure 2 (Left): $b$ shifts the peak of the gradient after which an exponential decay occurs. As $b$ increases only neurons with lesser importance are affected by the regularisation. With the weights of such neurons receding towards zero, the final distribution of importances will have a polarised bimodal shape - only important and zero-weight neurons will be left (Right).

## 5 EXPERIMENTS

In this section, we will evaluate our sparsity regularisation under a one-shot pruning setting on the CIFAR10/100 and ImageNet datasets using VGG and ResNet models. In most structured pruning literature evaluation is performed only under relatively benign pruning ratios, which mostly preserve the performance of the original model and can make it difficult to compare related methods. In our evaluations we observe that the difference between approaches becomes noticeable when pruning ratios are increased. Under more inauspicious circumstances we show that our method outperforms existing state-of-the-art approaches and can, in some instances, significantly mitigate the detrimental effects of severe pruning. We also compare the pruned VGG/ResNet networks with similarly sized MobileNetV2 models and show that pruning outperforms these compact architectures.

### 5.1 EXPERIMENTAL SETUP

**Pruning.** Under a one-shot pruning scenario, a model is trained, then pruned and subsequently fine-tuned to recuperate performance. An appropriate regularisation strength is selected such that it does not impede training while also producing enough sparsity to prune the desired amount of neurons. For practical reasons we train a model only once and then evaluate it under several pruning ratios. We prune networks in a global fashion, removing filters based on their importance score and without regard to the resulting distribution over the layers. To avoid the pruning of all neurons within a layer, a minimum of three filters will be preserved. While this avoids the extreme case of layer collapse, if a method disproportionately prunes the neurons of a single layer it could still irreparably damage the network. From a technical perspective, pruning the filters of a layer changes its parameter count but also that of the following layer, since the expected input dimensionality has changed. While this has no further implications for VGG style architectures, it does pose some challenges for networks with residual connections. Similar to related works, in order to avoid mismatched dimensions in ResNets, we do not prune skip connections or the last layer in residual connections.

**Related Methods.** We compare our approach with L1 Slimming (Liu et al., 2017), Polarization Regularisation (Zhuang et al., 2020), Deep Hoyer (Yang et al., 2019) and Uniform Channel Scaling (UCS) (Zhuang et al., 2020). The same experimental settings are used for all methods during training, pruning and fine-tuning. An exception is the UCS baseline, which does not use any sparsity regularisation and prunes in a local manner by removing the same percentage of filters across all layers. All networks of related approaches are built using vanilla BatchNorm, and with the same initialisation scheme as Liu et al. (2017). Models trained with $\sigma$BatchNorm layers are initialised with $\gamma$ drawn from a standard Normal distribution and will not use weight decay during training. The learning rate of $\sigma$BatchNorm is set to be higher by factor of 10 than that of the rest of the network - we found this to help with polarisation.

**Hyper-parameters.** The same settings as Liu et al. (2017) are used for the CIFAR10 and CIFAR100 datasets. VGG-16 and ResNet-56 models are both trained and fine-tuned for 160 epochs, with the same learning rate schedule and with a batch size of 64. Models are optimised using SGD with momentum, weight decay of $10^{-4}$ and an initial learning rate of $10^{-1}$ which reduces stepwise by

a factor of $10^{-1}$ at epochs 80 and 120. The respective sparsity regularisations are applied on all layers of the models. Datasets are augmented using only random crops and horizontal flips. On ImageNet, we evaluate methods on the ResNet-50 bottleneck architecture and regularise only its prunable layers. Models are trained for 90 epochs and fine-tuned for 30, using a batch size of 512. The initial learning rate is $10^{-1}$ for training and $10^{-2}$ for fine-tuning, and is scheduled to decrease after 1/3 and 2/3 of the respective training time. For our RNI experiments the following hyper-parameters are used: $\lambda_S = 10^{-3}, b = 3$ for VGG-16, $\lambda_S = 10^{-4}, b = 0$ for ResNet-56, and $\lambda_S = 10^{-5}, b = 3$ for ResNet-50.

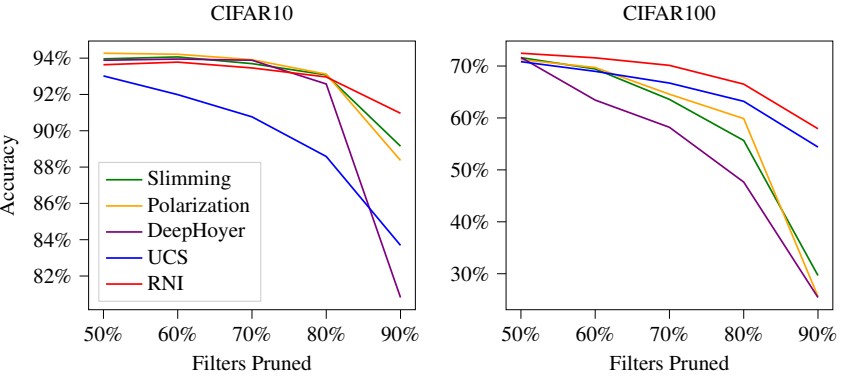

Figure 3: Accuracy of fine-tuned VGG-16 models at varying pruning thresholds. Performance differences between methods become more pronounced at higher ratios. RNI degrades gracefully compared to related methods.

## 5.2 RESULTS

Apart from ImageNet, all methods are evaluated on three random seeds and their mean performance is reported. For clarity, we will omit the standard deviation as we found the results to be very similar across runs. Two scenarios are examined in more detail as they sit on opposite ends of the difficulty spectrum: pruning 50% and 90% of filters. The 50% mark is the default setting in the structured pruning literature and we expect all methods to recuperate most of their baseline performance. This pruning level also ensures a fair comparison between methods as it acts as an anchor for hyper-parameter choices. At 90% of filters pruned, we see a significant degradation in performance and a far larger differentiation between methods and models.

In Figure 3 we have an overview of how VGG networks degrade as pruning ratios increase. On both datasets our method deteriorates gracefully compared to the abrupt declines of related approaches. While on CIFAR10 all methods have an easier time maintaining their performance, on CIFAR100 degradation becomes immediately visible due to the increased difficulty of the dataset. Surprisingly, the locally pruning UCS baseline performs worst on CIFAR10 but outperforms all methods, apart from ours, on CIFAR100. This shows that correctly identifying unimportant neurons in a global manner becomes harder as the difficulty of the dataset increases. In our ablation studies we will show that this deterioration can be directly linked to excessive pruning of early layers in VGGs.

**CIFAR10.** This task is relatively simple and we expect excess model capacity to be prunable without much performance loss, as seen in Table 2. The only significant degradation happens for VGGs at the 90% mark, where UCS and DeepHoyer lose over 10% accuracy compared to the baseline. ResNets are more robust and are lose less than 4% accuracy while reducing roughly 80% of FLOPs.

**CIAFR100.** Due to the increased difficulty, in Table 3 we see a larger deterioration for models trained on the CIFAR100 dataset. Here networks require more capacity, thus the number of necessary filters should be higher, and the identification of superfluous ones more difficult. With 90% of filters removed, VGG networks experience severe degradation as methods fail to identify which neurons to keep. In this scenario our approach considerably outperforms state-of-the-art methods by 30% classification accuracy. The only exception is the UCS baseline, which by design maintains the stability of a model, as it will never prune excessive amounts of filters in any given layer. ResNets again prove to be more robust to pruning and show less variation in performance between methods.

| VGG-16 | Baseline | 50% | | 90% | | Resnet-56 | Baseline | 50% | | 90% | |
|---|---|---|---|---|---|---|---|---|---|---|---|
| | Acc. | Acc. | FLOPs | Acc. | FLOPs | | Acc. | Acc. | FLOPs | Acc. | FLOPs |
| Slimming | 93.71 | 93.96 | 60.25 | 89.15 | 11.40 | Slimming | 93.68 | 93.45 | 54.07 | 89.90 | 19.55 |
| Polarization | 93.91 | 94.27 | 64.15 | 88.37 | 14.69 | Polarization | 93.34 | 93.50 | 52.69 | 89.43 | 18.89 |
| DeepHoyer | 93.66 | 93.96 | 61.32 | 80.82 | 8.09 | DeepHoyer | 93.44 | 93.51 | 51.07 | 89.94 | 20.84 |
| UCS | 93.84 | 93.02 | 25.19 | 83.69 | 1.13 | UCS | 93.78 | 93.33 | 50.39 | 88.78 | 14.79 |
| RNI | 93.53 | 93.64 | 48.24 | **90.96** | 8.40 | RNI | 93.53 | 93.01 | 56.13 | **90.26** | 21.54 |

Table 2: Results on CIFAR10 for different pruning ratios, showing absolute accuracy (%) and remaining FLOPs (%) relative to the unpruned baseline. Due to the simplicity of the task, models can be significantly pruned without deterioration.

| VGG-16 | Baseline | 50% | | 90% | | Resnet-56 | Baseline | 50% | | 90% | |
|---|---|---|---|---|---|---|---|---|---|---|---|
| | Acc. | Acc. | FLOPs | Acc. | FLOPs | | Acc. | Acc. | FLOPs | Acc. | FLOPs |
| Slimming | 73.54 | 71.61 | 61.12 | 29.66 | 3.76 | Slimming | 71.94 | 70.06 | 56.97 | 63.92 | 15.29 |
| Polarization | 73.47 | 71.23 | 63.82 | 25.62 | 2.59 | Polarization | 71.80 | 70.08 | 40.78 | 62.42 | 15.47 |
| DeepHoyer | 73.19 | 71.66 | 67.23 | 25.42 | 4.98 | DeepHoyer | 71.88 | 71.14 | 49.22 | 63.49 | 19.51 |
| UCS | 73.80 | 70.84 | 25.19 | 54.39 | 1.13 | UCS | 71.90 | 70.72 | 50.40 | 63.65 | 14.79 |
| RNI | 72.28 | 72.46 | 36.72 | **57.92** | 6.31 | RNI | 71.30 | 70.65 | 48.24 | **63.95** | 16.34 |

Table 3: Results on CIFAR100 for low and high ratios of filters pruned. Shown are the absolute classification accuracy (%) and remaining FLOPs (%) relative to the unpruned baseline. Due to the increased complexity of the classification task, it becomes more difficult to remove excess capacity. The VGG architecture breaks down significantly after sever pruning.

**ImageNet.** On ImageNet in Table 4 the results are similar, however we observe bigger losses due to the increased dataset difficulty. Under these circumstances our method still outperforms prior state-of-the-art, but is worse than the UCS benchmark. This again indicates that for more difficult datasets, global methods over-prune the wrong layers, something the UCS baseline will not do by design. Nevertheless, ResNets show remarkable robustness to pruning, which we attribute to the fact that their skip connections act as a fail-safe and limit the effect of over-pruned layers.

**Compact Networks.** At higher pruning ratios the parameter count of models is significantly reduced, raising the question of whether they would outperform networks specifically designed to be lightweight. Compact networks such as MobileNetV2 Sandler et al. (2018) have an adjustable width multiplier allowing the instantiation of models with arbitrary number of parameters. We thus conduct additional baseline experiments comparing the RNI pruned VGG/ResNet models from Tables 2 and 3 to non-sparse MobileNets of the same size. For each of the RNI pruned models we find a width multiplier such that the corresponding MobileNet has a similar

| ResNet-50 | Baseline | 50% | | 90% | |
|---|---|---|---|---|---|
| | Acc. | Acc. | FLOPs | Acc. | FLOPs |
| Slimming | 75.74 | 72.56 | 41.12 | 51.74 | 16.50 |
| Polarization | 75.12 | 73.11 | 46.12 | 49.43 | 17.00 |
| DeepHoyer | 75.36 | 68.72 | 57.56 | 50.98 | 24.14 |
| UCS | 76.14 | 73.28 | 44.78 | **58.13** | 16.94 |
| RNI | 74.75 | 72.31 | 48.13 | 54.08 | 17.66 |

Table 4: ImageNet results showing absolute accuracy (%) and remaining FLOPs (%) relative to the unpruned baseline.

number of parameters, and then optimise it using the same training setup, but without a sparsity loss. From the results in Table 5 we can see that our pruned models can outperform similar sized MobileNets in almost every setting, proving the practical utility of higher pruning ratios.

**Summary**. Because of the evaluation over datasets with varying degrees of complexity, we can make several observations about how models degrade under a spectrum of pruning ratios. Firstly, at 50% of filters pruned differences between approaches are barely visible since all methods are able to regain most of their pre-pruning performance. Differences become obvious only at higher ratios, where our method clearly surpasses prior state-of-the-art. Unsurprisingly, residual networks are more robust to pruning than VGG architectures, since their skip connections can propagate the signal even if essential residual connections have been mistakenly over-pruned. This however comes at the cost

| Network | Dataset | Pruning Ratio | | | |
|---|---|---|---|---|---|
| | | 50% | #P. | 90% | #P. |
| VGG-16 [RNI] | CIFAR10 | 93.64 | 3.40M | **90.96** | 167K |
| MobileNetV2 | CIFAR10 | 93.38 | 3.43M | 89.67 | 177K |
| VGG-16 [RNI] | CIFAR100 | 72.46 | 3.23M | **57.92** | 158K |
| MobileNetV2 | CIFAR100 | 75.16 | 3.12M | 52.72 | 163K |
| ResNet-56 [RNI] | CIFAR10 | 93.01 | 401K | **90.26** | 112K |
| MobileNetV2 | CIFAR10 | 91.21 | 408K | 88.55 | 124K |
| ResNet-56 [RNI] | CIFAR100 | 70.65 | 429K | **63.95** | 128K |
| MobileNetV2 | CIFAR100 | 71.05 | 452K | 52.72 | 163K |

Table 5: Comparison between non-sparse MobileNetV2 models and RNI pruned VGG/ResNets on CIFAR datasets. For each setting the width multiplier has been chosen such that the MobileNet will have the same number of parameters as the pruned networks.

of pruning efficiency, as seen by the diminished reduction of FLOPs compared to VGG models. To our surprise we find that while UCS is not suited for simple classification tasks, it outperforms prior global pruning methods on more complex datasets. Because of its design, the UCS baseline avoids any risk of excessively pruning individual layers, making models less prone to sudden performance deterioration. This leads to the conjecture that the other related methods over-prune certain layers, which leads to severe degradation in VGG models. In our ablative analysis we show that this is indeed the case, and make the realisation that our method is the only one which does not significantly prune early layers in a network. This offers an explanation why RNI noticeably outperforms similar approaches at high pruning ratios. Finally, from additional evaluations on compact architectures we see that pruned VGG/ResNet models outperform similarly sized MobileNets, which further proves the utility of pruning as a means to obtaining compressed networks.

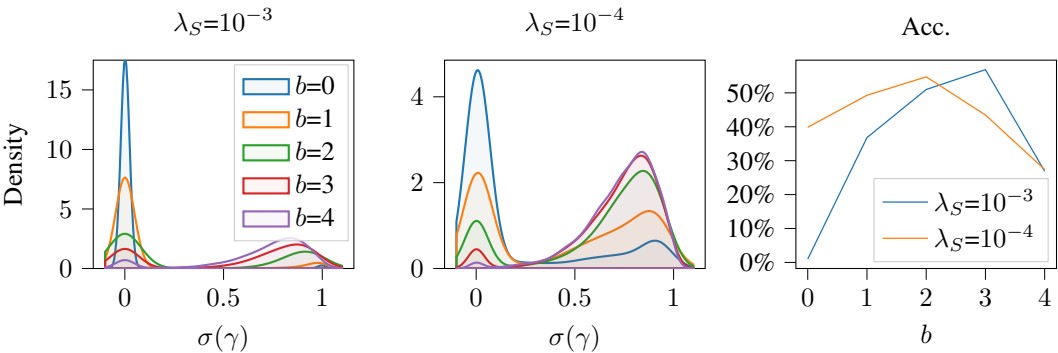

Figure 4: Effects of hyper-parameters $\lambda_S$ and $b$ for VGG-16 on CIFAR100. Left, Middle: Filter importance distributions after regularised training. Increasing $\lambda_S$ or lowering $b$ will produce sparser models. Right: One-shot performance at a 90% pruning ratio. Both over- and under-regularising can lead to performance degradation, however the search space for $b$ is limited.

## 5.3 ABLATIVE ANALYSIS

In this section we will perform two ablation studies to better understand the behaviour of the Receding Neuron Importance regulariser. We first analyse the effects of different hyper-parameters choices on neuron sparsity and one-shot pruning performance. Following that we investigate how much each layer of a VGG network will be pruned under comparable approaches.

**Sparsity.** In addition to the regularisation strength $\lambda_S$, our approach introduces the shifting parameter $b$. We show that these two parameters are complementary in controlling the gradient of the regularisation loss and effective at producing sparsity. While $\lambda_S$ scales the amplitude of the gradi-

ent, $b$ will set the range at which neuron importances start to recede. Figure 4 (Left, Middle) shows how $\lambda_S$ and $b$ choices create different distributions of neuron importances for a VGG model on CI-FAR100. Sparsity can be measured as the density around the zero importance limit, and increases with higher $\lambda_S$ or lower $b$. The interplay between these two parameters creates flexibility in the level of sparsity produced and the distribution of the remaining neuron importances. This ability to create a high variety of distributions comes at the cost of relative hyper-parameter sensitivity with respect to pruning accuracy, however this limitation only becomes visible at high pruning thresholds. Figure 4 (Right) shows the one-shot pruning accuracy at a 90% pruning ratio, of models trained with selected hyper-parameters. We can observe that both under- and over-regularised models will suffer from severe pruning. Unsurprisingly, inducing a strong degree of sparsity during training does not translate into a highly prunable model.

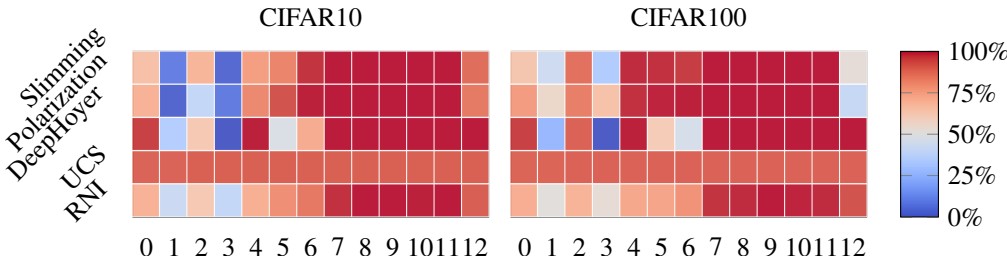

Figure 5: Percentage of filters removed from each layer when pruning a VGG-16 model at a 90% ratio. The deeper half of layers contain over 70% of the total amount of filters in the network. RNI stands out by not over-pruning layers in the first half.

**Pruning Locations.** Figure 5 shows what percentage of filters are removed from each layer during pruning. UCS prunes in a local manner, removing the same fraction of filters from each layer, as can be seen from the unchanging colour-coding. It is worth reminding that in a VGG-style architecture the number of filters periodically doubles as the network becomes deeper.This has the consequence that global pruning methods will achieve their target in a large part through the amount of filters pruned from the deeper layers. These are less sensitive to pruning, as there will still be a high absolute number of filters left. Consequently, removing a large percentage of neurons from the first layers will not add much to the global target, but will significantly limit their expressive power. Thus, it is reasonable to assume that filters in the early layers of a network have a higher importance, since the capacity of these layers is already limited.

From Figure 5 we can see that deeper layers are indeed pruned more heavily than ones closer to the input. There are however some exceptions: on the CIFAR10 dataset, the only method pruning a significant amount of filters from the first layer is DeepHoyer - which also achieves the lowest accuracy based on the results from Table 2. On the CIFAR100 dataset Network Slimming and Neuron Polarization start excessively pruning from the 5th layer onward. RNI is the only approach that starts to heavily prune only in the latter half of the network. Our method, along with UCS, are the only ones that do not suffer catastrophic damage after pruning 90% of the total number of filters.

## 6 CONCLUSION

In this work we introduce RNI, a novel sparsity regularisation for structured pruning, under which only the weights of unimportant neurons is designed to recede. To this end, we propose $\sigma$BatchNorm, a BatchNorm variation with bounded scaling factors, enabling the construction of such targeted regularisation functions without sacrificing performance. Our method consistently outperforms state-of-the-art methods for VGG/ResNet architectures on CIFAR and ImageNet datasets. We show that accurately comparing methods requires evaluation on a spectrum of pruning ratios, as differences become more pronounced at higher levels. For VGG architectures our method significantly reduces the performance degradation from severe pruning compared to prior art. In future work we would like to explore more applications of the RNI regulariser as well as a way to schedule its hyper-parameters to obtain a pre-determined sparsity ratio.

## REPRODUCIBILITY STATEMENT

This paper is entirely reproducible. All experimental details including setup and hyper-parameters are described in depth in Sections 5.1 and 5.2. All reported results have been run on publicly available datasets. Furthermore, a runnable PyTorch repository has been linked and attached as supplementary material.

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
