# OpenReview forum: "Receding Neuron Importances for Structured Pruning"
_ICLR.cc/2023/Conference — Submitted to ICLR 2023_

### Official Review · Reviewer_uAwb · 2022-10-24

**Confidence:** 3
**Correctness:** 3
**Technical Novelty And Significance:** 3
**Empirical Novelty And Significance:** 2
**Recommendation:** 6

**Clarity, Quality, Novelty And Reproducibility:**

- The idea to obtain scaling factors that enable better discrimination between unimportant and important neurons is well-motivated and intuitive.
- The paper is clearly written and well polished. I think most members of the community would be able to easily dissect it without requiring substantial prior knowledge and without being misled.


**Strength And Weaknesses:**

Strength
- The idea that shrinks only the outputs of unimportant neurons while maintaining the others is well-motivated.
- The sigmoid batchnorm is sufficiently simple for people to build on, although the proposed regularization loss involves additional hyperparameters.
- I believe the experimental validation is quite strong and extensive; various architectures with different capacities were used, and the baselines are recently proposed methods and competitive.
- The authors provide detailed analyses on the hyperparameters and where to be pruned, which help understanding the network behavior.
- The paper is generally quite thorough and generally feels complete: I believe it is generally ready for publication if it is decided by the reviewers that significance is sufficient to warrant publication.

Weakness
- As the authors described, this paper and Polarization Regularizer (Zhuang et al., 2020) share the same motivation and take a similar approach in terms of optimizing batchnorm scaling factors in a desired way. Although the authors briefly mentioned the difference in the related work section (i.e., While the method effectively increases the margin between important and unimportant neurons, it does so by both shrinking and expanding weights.), I am not sure why the proposed method should be better than Polarization Regularizer. Could the authors provide the merits of your work over Polarization Regularizer in more detail?
- I am not sure whether the proposed method is effective for ResNets and is generally applicable to resblock-equipped networks. For instance, in the right part of Table 2, RNI slightly outperforms the baselines but with more FLOPs at 90% pruning ratio. In the right part of Table 3, at 90% pruning ratio, the accuracies are very similar. Furthermore, in Table 4, the superiority of RNI is more questionable: Slimming and Polarization yield better results with less FLOPs at 50% ratio.
- I think the comparison in Table 5 should be improved; matching the number of parameters may be too naive. I would recommend the authors to add FLOPs to Table 5 and perform additional experiments using the network with similar FLOPs or latency.


**Summary Of The Paper:**

This paper proposes a simple variation of batch normalization with a bounded scaling parameter to estimate the importance of each channel. To only suppress unimportant units while preserving important units, the authors design a regularization loss on these bounded scaling factors, with additional hyperparameters. The proposed approach is validated on CIFAR and ImageNet with VGG, ResNet, and MobileNet models.

**Summary Of The Review:**

In general, I believe this work did a good job in terms of designing interesting sparsity-inducing regularization for structured pruning. Although the in-depth comparison with Polarization Regularizer is necessary and some results are questionable, the experiments and analyses are quite good and thorough.

---

> ### Author Response · Authors · 2022-11-11
> **Reply to uAwb**
>
> We thank the reviewer for reading our work in detail and providing insightful comments.
>
> 1.**Comparison to Polarization Regularization**. As pointed out, our work shares similarities with Polarization Regularization, as both methods are motivated by the shortcomings of L1's indiscriminate parameter shrinking. However, the difference is in how this is achieved and what parameters are being targeted. The Polarization regulariser targets all scaling parameters by explicitly maximising the dispersion of their distribution. This is done through the addition of a penalty term based on the distance to the mean of scaling weights. This term is added to the L1 regulariser and differs with the behaviour of our contribution in two ways: 1) as with Slimming this still affects neurons deemed important by artificially increasing their weight, potentially interfering with the gradients from the classification loss. 2) It adds an inter-dependency between BatchNorm parameters as it requires the computation of the average to maximise dispersion.
>
>
> In contrast, our method shrinks only scalers with small magnitudes and does not provide any gradients if these are larger, hence not intefering with the classification loss. Furthermore, our method computes the regularisation strength of a scaler independent from other parameters. This not only is computationally more efficient but allows for more flexibility during training since it poses less constraints. For example, initially important filters could potentially be pruned later if their utility for classification decreases over time - this wouldn't be possible with Polarization as their value would be artificially kept high. If necessary, and constrained by space, we can further expand on this in the Appendix.
>
> 2.**ResNet Pruning Efficiency**. We agree that results for ResNets are less conclusive than those for VGGs. In section 5.2 we argue that residual networks are more robust to pruning due to the skip connections always maintaining the propagation of signal. Unlike VGGs if a layer is over-pruned it won't lead to drastic drops in performance. However, this also makes them harder to compress and poses additional restrictions on pruning, due to potential size mismatches of activations. In terms of FLOPs we see that results vary depending on the location of pruned channels, which does not have a clear relationship to accuracy and thus makes comparisons more difficult. Overall we believe experiments and results for ResNets can be improved not necessarily but changing the regularisation, but by developing new pruning strategies specialised for these architectures.
>
> 3.**Matching by Flops**
> Unfortunately matching methods by FLOPs is challenging as we can compute this metric only after pruning. The variable we can control is the amount of neurons to be pruned and we use this to enable a fair comparison between methods. Furthermore, in Table 5 matching ResNets with Mobilenets based on FLOPs is not really feasible, as the latter are very efficient due to the depth-wise separable convolutions.

---

> > ### Author Response · Authors · 2022-11-17
> > **Follow-up questions**
> >
> > We again thank for the feedback and would like to draw attention to the discussion period ending soon.
> > Hopefully our response has addressed all concerns and provided enough clarification to better evaluate our paper.
> > We are happy to answer any additional questions and appreciate any more feedback to further improve our work.

---

### Official Review · Reviewer_SWo9 · 2022-10-25

**Confidence:** 4
**Correctness:** 3
**Technical Novelty And Significance:** 2
**Empirical Novelty And Significance:** 2
**Recommendation:** 3

**Clarity, Quality, Novelty And Reproducibility:**

The paper is clearly written and easy to follow and the quality is good.

Novelty is limited. The original idea of regularizing BatchNorm parameters for channel pruning is proposed in Slimmable Network (Liu et al., 2017). Based on that, this paper proposed the adjustment for BatchNorm, which seems marginal and not well-supported by either mathmatical proof or strong experimental results.

**Strength And Weaknesses:**

Channel pruning is an important task in neural network compression and the effort towards better channel pruning methods is appreciable.

Experimental results are weak. The results on the ImageNet dataset are much worse than the previous state-of-the-art results.

[ref1] Soft filter pruning for accelerating deep convolutional neural networks

[ref2] Pruning filter via geometric median for deep convolutional neural networks acceleration.

[ref3] Metapruning: Meta-learning for automatic neural network channel pruning


**Summary Of The Paper:**

The authors proposed a channel-pruning method by regularizing the BatchNorm scaling parameters, and proposed a BatchNorm variation suitable for pruning. The authors point out that the previous BatchNorm-based pruning methods shrink all parameters with an equal gradient irrespective of their importance. Thus, the authors propose to remove the offset $\beta$ in BatchNorm and bound the scale parameter $\gamma$ by applying the sigmoid function before multiplication, and the Receding Neuron Importance regularization for pruning. However, the design is lacking in theoretical support and the authors didn’t prove the proposed method achieved better accuracy than previous methods.

**Summary Of The Review:**

The paper is well-written, however, the proposed method seems incremental, the design is lacking in theoretical support and the experimental results are weak.

---

> ### Author Response · Authors · 2022-11-11
> **Reply to SWo9**
>
> We thank the reviewer for taking the time to read our work and provide feedback.
>
> We would like to point out that the experimental setup used in our work is more rigorous than what is commonly done in the literature, and **we prune at much higher levels than the referenced works**. We make the case that in order to comprehensively compare methods one has to evaluate multiple pruning thresholds, as differences become apparent only in more difficult scenarios. Low thresholds do not accurately reflect the capabilities of a method, since the original performance is mostly preserved and any differences between approaches are within the margin or error. This can be seen throughout the experimental results (Tables 2-4) as all evaluated state-of-the-art methods perform similarly at a 50\% pruning ratio. Our experimental evaluation is extensive as it includes datasets of increasing difficulty and a wide range of pruning thresholds. At the high end of the pruning spectrum we remove 90\% of neurons which will inevitably lead to much lower performance. At low end of the spectrum we prune 50\% of neurons, which is still higher than that used in the works referenced in the review. The pruning ratios of referenced works are: [ref1] 30\%; [ref2] 30-40\%; [ref3] 15-50\%.
>
> In comparison with related state-of-the-art methods, our work shows strong experimental results. We outperform sparsity regularised pruning methods in all evaluated scenarios and significantly reduce the impact of severe pruning regimes (+28\% VGG16/CIFAR100, +2\% ResNet50/ImageNet). We believe that this type of regularisation is an important line of research as it provides a simple way of analysing filter importances, as well as being widely applicable due to the prevalence of BatchNorm layers.

---

> > ### Author Response · Authors · 2022-11-17
> > **Follow-up questions**
> >
> > We again thank for the feedback and would like to draw attention to the discussion period ending soon.
> > Hopefully our response has addressed all concerns and provided enough clarification to better evaluate our paper.
> > We are happy to answer any additional questions and appreciate any more feedback to further improve our work.

---

### Official Review · Reviewer_LW7m · 2022-10-29

**Confidence:** 4
**Correctness:** 3
**Technical Novelty And Significance:** 3
**Empirical Novelty And Significance:** 3
**Recommendation:** 5

**Clarity, Quality, Novelty And Reproducibility:**

The paper is clearly written. I have not tried to run the provided code to reproduce the results.

**Strength And Weaknesses:**

Strengths:
1. The proposed measure and regularizer seem to be a more principled magnitude based pruning method than existing methods that directly work on the magnitude and apply an L1 regularizer.

2. The results on CIFAR-10, 100 suggest that indeed this method has potential, since it outperforms existing methods on higher pruning ratios.

Concerns:
1. The proposed method is only tested on 2 architectures: ResNet and VGG, and on image tasks CIFAR and ImageNet.

- Can the method be applied to Transformer architecture on language tasks?
- The paper applies the method only for one-shot pruning. Can this be applied in multiple iterations, pruning a certain fraction of neurons in each round?
- Can this method be used to prune pre-trained networks? For example, take a pretrained network and fine-tune if for a few epochs on a downstream task, and then apply this pruning method.

2. Although the method performs well on simpler tasks like CIFAR, it performs worse than UCS on difficult tasks like ImageNet. The paper says that this is because "This again indicates that for more difficult datasets, global methods over-prune the wrong layers". I think this is a major limitation. However, I think this might be remedied, for example, by a per-layer re-normalization of the importance scores, so that a similar ratio of neurons are pruned in every layer.

3. (Minor) The paper promotes the proposed method by claiming that it measures the 'importance' of neurons by bounding the scaling parameter of batchnorm in the range [0, 1]. However, I am not fully convinced by this. Consider the case when a neuron has a low scaling parameter, but in its outgoing weights to the next layer, a few (say 10%) of the weights are large. That means that even though the output magnitude of that neuron is low, some neurons in the next layer give high weightage to its output. In that case, this neuron can still be important. In short, what I mean to say is that just the magnitude of a neuron's output might not necessarily capture its 'importance'. While this method still seems more principled than previous methods at using the scaling parameter's magnitude for pruning, I do not agree that it captures the importance of the neuron. I would suggest clarifying this in the paper.


**Summary Of The Paper:**

The paper proposes a new way to measure the 'importance' of a neuron, by re-parameterizing the scaling parameters of the batchnorm layers. The proposed method is to use a sigmoid on the scaling parameter, to ensure that it lies in [0,1], and the value of the sigmoid reflects the importance. Using this value (value closer to 1 means important), the 'unimportant' neurons are pruned in a one-shot manner. The paper also proposes a new regularizer for this scaling parameter, to control the amount of neurons to be pruned. The results on simpler image classification tasks like CIFAR-10, -100 suggest that this method indeed outperforms other methods, particularly at higher pruning levels. However, on more complex task like Imagenet, it seems to perform worse than the UCS method. Further, the algorithm is only evaluated on image tasks (CIFAR and ImageNet), and two architectures (ResNet and VGG).

**Summary Of The Review:**

I think the proposed method to measure the 'importance' of a neuron is more principled than previous methods that use magnitude, but I think the method can be more refined to deal with the problem of over-pruning the wrong layers for complex tasks, and by adding more experiments to show how the method performs for different architectures and tasks.

---

> ### Author Response · Authors · 2022-11-11
> **Reply to LW7m**
>
> We thank the reviewer for the constructive feedback and valuable suggestions for further applications.
>
> 1.**Scope and benchmarks**: The focus of this paper is to motivate our novel contribution and comprehensively compare it by adhering to a rigorous, reproducible benchmarking setup consistent with prior works. To this end we chose to conform to the standards set in the relevant literature, which happens to be centred around VGG/ResNet architectures, the mentioned image datasets and a one-shot, global pruning scenario. The scope of our evaluation is in fact larger than that of previous works in two ways: 1) we evaluate on both CIFAR datasets to assess the efficacy of global pruning methods as datasets become progressively more difficult; 2) we evaluate several pruning ratios and make the case that this is a necessary requirement for accurate comparisons - rather than the common practice of pruning at a single, benign threshold. Extending this setup for other application domains or pruning strategies is indeed interesting, however it is beyond the scope of this paper. Exploring compression of pre-trained Transformers or mitigating fundamental shortcomings of global pruning strategies are applications not restricted to our contribution, but which we can nevertheless tackle in future work.
>
> 2.**UCS vs global pruning**. Our contribution falls in the category of structured pruning using sparsity regularisation on BatchNorm layers. As such, our main comparison is to prior works of the same category (Slimming, Polarization Regularisation) or ones that have a similar motivation in the design of the regulariser (DeepHoyer). To stay consistent with the literature we evaluate these methods in their native pruning setting: one-shot global pruning. For completeness we also included a simple local pruning baseline (UCS) which does not use any sparsity regularisation. Because of our extensive evaluation we were able to make the surprising observation that the efficacy of global importance ordering decreases relative to this baseline - only in settings where the pruning ratio is larger than the spare capacity (Figure 3). This observation holds for all examined global pruning methods, and is not restricted to our contribution. As suggested, we can address these limitations by shifting our focus from sparsity regularisers to pruning strategies: either through layerwise re-normalisation or iterative pruning. Because such improvements could be applied to any regularisation based method it is beyond the scope of this paper. In the standard global setting we do show that our approach consistently performs best among similar regularisation methods and significantly reduces the deterioration caused by high pruning ratios (+28\% VGG16/CIFAR100, +2\% ResNet50/ImageNet).
>
> 3.**BatchNorm based neuron importance**: We agree that scenarios can be constructed where small BatchNorm scalers are reversed by large weights in the filters of subsequent layers. However, this is something that will not happen in practice due to the configuration of the training setup, in particular: the use of the sparsity regularisation on the scalers together with the L2 penalty imposed on filter weights. We designed our regulariser to increase its strength the closer values are to 0, which leads to BatchNorm weights that are infinitesimal. In order for subsequent layers to revers that, their filter weights would have to be several orders of magnitude larger - something impossible to achieve while being subject to an L2 penalty. We do mention weight decay in our experimental setup, but as per the suggestion we can further emphasise its relation to neuron importances.

---

> > ### Author Response · Authors · 2022-11-17
> > **Follow-up questions**
> >
> > We again thank for the feedback and would like to draw attention to the discussion period ending soon.
> > Hopefully our response has addressed all concerns and provided enough clarification to better evaluate our paper.
> > We are happy to answer any additional questions and appreciate any more feedback to further improve our work.

---

### Decision · Program_Chairs · 2023-01-20

**Decision:**

Reject

**Justification For Why Not Higher Score:**

The evaluation does not support the claim that this method is an improvement in the most challenging settings.

**Justification For Why Not Lower Score:**

N/A

**Metareview: Summary, Strengths And Weaknesses:**

**Summary:** This method proposes a slight tweak to the way we do channel pruning in convolutional neural networks. Rather than pruning based on the batchnorm gamma parameter, it adds a sigmoid around that parameter (both in the network and for pruning purposes) and prunes based on that. This ensures that all of these parameters are within the same scale range. The authors claim that the method "significantly outperforms existing approaches," but the evaluation doesn't seem to support that claim.

**Strengths:** The modification is simple and easy to implement.

**Weaknesses:**
* This method doesn't conclusively do better than other methods. The authors tout the smallest-scale experiments on VGG-like networks for CIFAR, but the method is not definitively better than others on ResNet-50 on ImageNet (which is the de facto standard task in pruning right now). Several of the comparisons are at very different FLOP numbers and accuracies for different methods, so it's difficult to make an apples-to-apples comparison. Pareto curves showing the tradeoff at many different points for these methods would be more helpful. Despite the authors' claim that this method "significantly outperforms existing approaches," that is not apparent from the results. There are other methodological weaknesses in the paper, such as a lack of error bars, that suggest the evaluation is incomplete.
* The method is really simple. There isn't much novelty. A single sigmoid is added in one place. No, a method doesn't have to be complicated to be effective (and many of the best tricks we have in deep learning are very simple), but this method (a) is a minimal change, (b) isn't clearly justified, and (c) doesn't actually help very much. This is in contrast to similarly small changes like residual connections, batchnorm, or dropout, which are small but very significant in their impact on the resulting network.

**Recommendation:** Reject

**Summary Of Ac-Reviewer Meeting:**

N/A